# Positive Well-Being, Work-Related Rumination and Work Engagement among Chinese University Logistics Staff

**DOI:** 10.3390/bs14010065

**Published:** 2024-01-18

**Authors:** Siyao Zheng, Shuyue Tan, Xiaotong Tan, Jialin Fan

**Affiliations:** 1School of Psychology, Shenzhen University, Shenzhen 518000, China; 2021125855@email.szu.edu.cn (S.Z.); tessshuyue.work@gmail.com (S.T.); 2110481044@email.szu.edu.cn (X.T.); 2The Shenzhen Humanities & Social Sciences Key Research Bases of the Center for Mental Health, Shenzhen 518060, China

**Keywords:** well-being, satisfaction, happiness, work-related rumination, work engagement, university logistics staffs, mental health

## Abstract

Logistics personnel in Chinese universities are facing unbalanced costs and benefit from overloaded work with minimum wages, which impede school development and their well-being. However, the logistics staff population has been neglected in past investigations pertaining to psychological health conditions. The present study aimed to examine the positive well-being, work-related rumination, and work engagement of logistics staff, their correlations, and the factors affecting well-being in 282 Chinese university logistics staff via the Smith Well-being Questionnaire, the Work-Related Rumination Questionnaire, and the Utrecht Work Engagement Scale. The results indicated low levels of well-being and high levels of work-related rumination and work engagement among Chinese university logistics staff. The presence of positive attitudes towards life and work and high levels of work engagement predicts enhanced well-being, while the presence of negative characteristics and work-related rumination predicts decreased well-being. In situations where the working hours and work duties are challenging to change, universities can regularly schedule psychological counselling sessions for logistics staff to improve their well-being.

## 1. Introduction

In November 2022, an unforeseen self-destructive episode garnered widespread attention across Chinese social media platforms. The individual implicated in this occurrence was from the logistics personnel within a Chinese higher education institution [1]. While the underlying factors that precipitated this incident remain intricate and contentious, it underscored the imperative for investigations into the occupational milieu and psychological well-being of logistics staff, as well as the implementation of preventative measures to avert comparable incidents in the future, which had been neglected in past decades. Logistics personnel hold a crucial position within the school framework and play a pivotal role in advancing higher education’s comprehensive development [2]. The high levels of working stress and multifaceted work environments they constantly confront may result in diminished well-being, either at work or outside of work. However, while prior research predominantly concentrated on the mental health of academic staff [3] and other administrative staff [4] in the university, the well-being of logistics staff was neglected.

Mental health issues, such as depression and anxiety, are common in the Chinese working population [5]. A variety of Chinese employees have low wages and a lack of effective long-term occupational health services, while they chronically experience high workloads [6], resulting in an increased risk for psychological and physical fatigue and sleep problems [7]. Overtime work among Chinese employees is common. The average work hours of Chinese employees in 2017 was 60.73 h per week, which exceeded the Chinese Labor Law legal standard of work hours (i.e., 44 h per week) by 38% [7]. According to previous studies, shift work, long working hours, and hazardous work environments (e.g., exposure to noise or toxic chemicals) are also potential risk factors for the poor mental health of Chinese workers [8,9,10].

Logistics staff in Chinese universities play a multifaceted role, including dining services management, student housing, facility maintenance, procurement, and inventory control. Based on the nature of their work, university logistics staff can be categorized into technical employees (e.g., maintenance service personnel, procurement and inventory managers, and utility managers) and non-technical employees (e.g., dining hall staff and student housing administrators). Their working environments are usually accompanied by various ergonomic hazards, such as heavy lifting, repetitive movements, and prolonged standing, which significantly impact employees’ well-being [11,12]. In addition, the shortage of human resources results in long working hours and a heavy workload for these staff. Chinese university logistics staff wages have consistently remained around the minimum wage in China (1420–1780 CNY/month) [13], while the per capita disposable income of Chinese residents is about CNY 3623 per month. This underscores a growing contradiction between the levels of socioeconomic development and the demands of people’s living standards.

The problems Chinese logistics staff face can result in low work engagement. Work engagement is a positive and fulfilling work-related state of mind characterized by vigorousness, dedication, and absorption [14]. Vigorousness involves high energy, mental resilience, effort investment, and perseverance in the face of difficulties. Dedication is perceived as a sense of significance, enthusiasm, inspiration, pride, and challenge at work. Absorption refers to being fully concentrated and happily engrossed in work, making it difficult to detach oneself from the tasks [15]. Remuneration has a significant positive effect on work engagement [16]. Job demands such as workload, contribute to a decline in individuals’ connection and passion for their work, ultimately resulting in a negative impact on their overall work engagement [17]. Chinese university logistics staff are generally dissatisfied with their salaries due to the incongruence between their wages and the workload [2]. Therefore, it is possible to observe a decrease in work engagement in Chinese university logistics staff.

The imbalance between salary and workload can lead to increased work-related rumination. Work-related rumination is a state in which one constantly thinks and ruminates about work issues outside of work [18]. It can be divided into two dimensions, affective rumination and problem-solving pondering [19]. Affective rumination is a repetitive thinking process that directs attention toward distress symptoms and associated feelings [20], resulting in a negative emotional response. Problem-solving pondering is unemotional, prolonged thinking about solutions to particular work-related problems [21], whose consequences may be positive if they result in a solution. Worldwide, at least 70% of employees experience work-related rumination, with this number gradually increasing [22]. A demanding and stressful work environment, requiring increased cognitive and emotional processing and setting higher goals, can induce rumination, especially in workers within the service sector and knowledge-intensive occupations [22,23]. Given the high cognitive and emotional demands associated with the management and communication tasks of Chinese university logistics staff, they are likely to be absorbed in either the affective aspects or the practical issues regarding their work outside of work.

Work rumination is negatively correlated with work engagement [24]. The presence of affective rumination and problem-solving pondering can act as a negative predictor of work engagement two years later [19]. Given the goal-directed nature of problem-solving pondering, employees may engage in problem-solving pondering during off-job hours, indicating a high level of work engagement [25].

Well-being is crucial. It is defined as how one feels, either in personal daily life or at work, and how one evaluates one’s life as a whole [26]. Individuals with positive well-being enjoy better physical health, greater accomplishment, better social relationships, and more productive economic contributions to society [27,28,29]. The well-being of employees can predict key indicators of organizational performance [30,31], such as productivity [32,33], absenteeism [34], job performance [35], and voluntary turnover [36]. In China, people who have higher incomes and employment are happier [37]. Therefore, it is necessary to explore the well-being of Chinese university logistics staff, given their relatively lower income and higher workload.

Research on the associations between work-related rumination, work engagement, and well-being is scarce. Previous research on well-being at work has focused on job characteristics (e.g., job demands, job support/control) and individual characteristics (e.g., personality, healthy lifestyle), which were proposed in the Demands, Resources, and Individual Effects (DRIVE) model [38]. Positive characteristics (e.g., positive personality [39], job support and control [40], healthy lifestyle [41]) predict positive well-being. In turn, negative job characteristics, such as high job demands and a poor working environment, are associated with impaired well-being either at work or in family life [42]. However, it remains unknown if previous findings could be reflective of the logistics staff in Chinese higher education institutions who might be experiencing an unbalanced workload and income.

In summary, Chinese university logistics staff may be experiencing multiple challenges at work or outside of work. However, their working condition and mental health have been neglected in past decades. In light of these, the aims of the present study are threefold: (1) to examine the level of work engagement, work-related rumination, and well-being of Chinese university logistics staff at work and outside of work, (2) to measure their correlations, and (3) to identify the risk factors for the well-being of Chinese university logistics staff. We hypothesized that Chinese university logistics staff are experiencing high levels of work rumination and low levels of work engagement and well-being, with correlations that are consistent with the pertinent findings in other populations.

## 2. Materials and Methods

### 2.1. Ethical Approval

This study has been reviewed and approved by the Medical Ethics Committee at Shenzhen University. Informed consent was given to and signed by all participants.

### 2.2. Procedure

The survey was conducted online and offline from 15 April to 31 October 2023. Questionnaires, comprising SWELL, WRRQ, and UWES, were distributed through a mini program called “Wenjuanxing” on WeChat, as well as in person at different logistics workplaces in universities, such as cafeterias, sports facilities, and dormitory areas.

A brief statement of the purpose of the survey and requirements for completion was conveyed before participants filled out the questionnaire. Participants were instructed to complete the questionnaire within 15 min. The questionnaire began by providing participants with an informed consent form which stated that participation was voluntary. Participants were free to withdraw from the survey at any point and not respond to those questions that they felt uncomfortable answering. The participants were informed that the data were anonymous and confidential. The participants then completed the questionnaires including SWELL, WRRQ, and UWES-9. On completion, the participants were shown a debriefing statement that repeated the aims of the study and thanked them for their participation.

### 2.3. Participants

Participants were recruited from the logistics department across universities in different cities in China, among which 282 people completed the questionnaires. The response rate was approximately 99.3%. Among the effective respondents, there were 138 males and 144 females, ranging from age 16 to 76 years (M = 36.64, SD = 7.05). The participants carried out a range of jobs, including dining hall staff; student housing administrators; and maintenance service personnel.

### 2.4. Materials

The survey included questions about demographics, the Smith Well-being (SWELL) Questionnaire, the Work-Related Rumination Questionnaire (WRRQ), and the Utrecht Work Engagement Scale (UWES-9). Demographic information included gender, age, education level, marital status, number of children, employment status, daily work schedules, job types, and occupational level.

#### 2.4.1. Smith Well-Being (SWELL) Questionnaire

The Smith Well-being Questionnaire (SWELL), a single-item measurement using one item for one characteristic of well-being, was adopted to assess the well-being of the participants [40]. This scale has been used to study the well-being of the business process outsourcing industry [40] and railway staff [41]. The items in SWELL were derived from the Well-being Process Questionnaire (WPQ), which is based on the Demands–Resources–Individual Effects (DRIVE).

The SWELL questionnaire consists of 18 questions, which are separated into two sections. Each question is rated on a 10-point scale. The first section measured respondents’ individual characteristics (health-related behaviours, personality) and work characteristics (job demands, job control and support, levels of noise and fumes). The second section measures positive well-being (i.e., job satisfaction, happiness at work, life satisfaction, and happiness outside of work) and work–life balance. The variables in SWELL were dichotomized into high and low groups by using the thresholds, with variables above the thresholds categorized into the high group. Most variables were categorized into high or low groups (e.g., healthy and unhealthy behaviours, positive and negative personality, high and low job demands) using a threshold of 7, where scores above 7 were classified as high, and scores equal to or below 7 were classified as low. A median split was used to recode some variables whose scores were significantly above or below 7 (e.g., when 80% of participants’ scores on the variable were higher than 7). Previous researchers have compared single-item and multiple items (WPQ) and have confirmed the validity and reliability of this single-item measure of well-being [42]. Positive well-being characteristics (e.g., job control/support, job satisfaction, and happiness at work) have a Cronbach’s alpha of 0.81, while negative well-being characteristics (e.g., job demands, work-related stress, and anxiety/depression) demonstrate a Cronbach’s alpha of 0.65 [40].

Both forward and reverse translations were used to translate SWELL into Chinese. The Chinese version of SWELL has been used in the measurement of the academic fatigue of university students [43], the workload and fatigue of doctors and nurses in psychiatric hospitals [44], and the occupational fatigue and well-being at work of employees of a tech giant in China [45].

#### 2.4.2. Work-Related Rumination Questionnaire (WRRQ)

The Work-Related Rumination Questionnaire (WRRQ) [18] comprises two subscales: affective rumination and problem-solving pondering. Each subscale consists of five items, rated on a five-point scale (1 = “rarely or never”, 5 = “very often or always”). The WRRQ has been shown to have good reliability and validity and has been successfully used in several previous studies [25,46].

#### 2.4.3. Utrecht Work Engagement Scale (UWES-9)

Work engagement was measured with the Utrecht Work Engagement Scale (UWES-9) [47]. UWES-9 is a nine-item self-report scale grouped into three subscales (vigour, dedication, and absorption), with each subscale consisting of three questions. All items are scored on a 7-point scale, measuring the frequency from 0 (never) to 6 (always). A cross-cultural study reported the high internal consistency reliability (Cronbach’s α) of the UWES-9 (α = 0.900) and its subscales —vigour (α = 0.808), dedication (α = 0.819), and absorption (α = 0.729). The validity of UWES-9 was also confirmed [47].

### 2.5. Statistical Analyses

Data analysis was carried out using SPSS 26. Descriptive statistics present the demographic information of university logistics staff. The frequency of responses for each variable in the SWELL questionnaire was analysed descriptively. Independent *t*-tests and ANOVA were conducted to compare the results of the Work-Related Rumination Questionnaire and Utrecht Work Engagement Scale with different demographic factors. Correlations between the variables in SWELL, work-related rumination, and work engagement were examined using Pearson correlation tests. Finally, logistic regressions were performed to uncover the factors that predict the positive well-being (including life satisfaction, happiness, job satisfaction, and happiness at work) of the participants.

## 3. Results

### 3.1. Descriptive Statistics

The descriptive statistics of education, marital status, number of children, employment status, daily work schedules, job types, and occupational level are listed in Table 1.

The variables in SWELL were dichotomized into high and low groups by the thresholds of the score of seven or the median, with variables above the thresholds categorized into the high group.

The descriptive statistics for each variable in the SWELL questionnaire are shown in Table 2. Over 70% of participants had scores above the threshold on health-related behaviours and personality. However, less than 35% of participants had scores above the threshold on life stress, anxiety/depression, musculo-skeletal problems, and exposure to fumes. The sample showed that satisfaction and happiness are low. In addition, well-being outside of work (life satisfaction, happiness) is higher than that at work (job satisfaction, happiness at work).

The mean score of WRRQ was 28.73 ± 4.80, with the score of affective rumination higher than that of problem-solving pondering. The total score of UWES-9 of university logistics staff was 36.21 ± 12.51, in which the vigour dimension scored the highest and the dedication dimension scored the lowest. The data are shown in Table 3.

### 3.2. Correlations between Positive Well-Being, Work-Related Rumination, and Work Engagement

Bivariate correlations between the observed variables are presented in Table 4. Well-being at work and outside of work (including life satisfaction, happiness, job satisfaction, and happiness at work) was significantly and positively related to health-related behaviours, personality, job control and support, and work engagement. It was significantly and negatively associated with noise and vibration, fumes, job demands, perceived stress at work, physical and mental fatigue, and work-related rumination (*p* < 0.05).

### 3.3. Comparative Study of Work-Related Rumination and Work Engagement Based on Different Factors

Independent samples t-test and ANOVA were conducted. The results (Table 5) showed a significant difference in work-related rumination among logistics staff of different age groups, education levels, marital status, daily work schedules, and job types (*p* < 0.05). The results presented the differences in work engagement across different occupational levels among university logistics staff (*p* < 0.05). The post hoc comparison data for factors with significant differences are presented in Table 6.

### 3.4. Predictors of Well-Being

Logistic regressions were conducted to investigate the predictors of low/high well-being. The dependent variables were high or low levels of life satisfaction, happiness, job satisfaction, and happiness at work. The independent variables were either categorical or continuous. Categorical independent variables were health-related behaviours, personality, life stress, anxious/depressed, noise and vibration, fumes, job demands, job control and support, perceived stress at work, physical and mental fatigue, efficiency at work, work–life balance, and anxious/depressed because of work from the SWELL questionnaire which were labelled with “high” if they scored higher than the threshold (i.e., seven points) and with “low” if scored lower than seven. Musculo-skeletal problems, with 87.9% of scores at or below seven, was dichotomized using a median of three. The predictors were input into the model together simultaneously. The Odds Ratio (OR) effect size for each of the independent variables (IV) is shown in Table 7, Table 8, Table 9 and Table 10 below.

## 4. Discussion

The present study examined the levels of well-being, work-related rumination, and work engagement of Chinese university logistics staff. The well-being of the participants is assessed via four variables (i.e., life satisfaction, happiness, job satisfaction, and happiness at work) in the SWELL questionnaire. The level of work-related rumination and work engagement are assessed via the total scores of the WRRQ and UWES-9.

The participants’ reports on well-being and work-related rumination are consistent with our hypotheses. Participants reported low levels of well-being, with well-being outside of work being higher than at work. In non-working time, logistics staff tended to ruminate more about the affective aspects of their work than the practical aspects, indicating that university logistics staff experience a high frequency of negative emotions, tension, and frustration in their work. However, a relatively high level of work engagement was presented, reflecting positive and active involvement in their work and responsibilities, which is inconsistent with the hypotheses.

The bivariate correlation results support our hypotheses, indicating a positive association between well-being and work engagement and positive characteristics (health-related behaviours, personality, and job control and support), and a negative association with work-related rumination and negative characteristics (noise and vibration, fumes, job demands, perceived stress at work, and physical and mental fatigue).

The logistic regression results, consistent with the hypothesis, suggest that low work-related rumination and high work engagement predict positive well-being. The results of logistic regression are also in line with the work of the previous researchers using other professional group samples, such as academics [48] and business outsourcing staff [40], which confirms that positive characteristics predict positive well-being and negative job characteristics are damaging to well-being at work. As previous studies have shown [40,41], the results showed that a healthy lifestyle, an optimistic personality, and high levels of job support/control increased well-being at work and outside of work. Anxiety and depression outside of work resulted in lower well-being. It was also found that high efficiency at work predicted positive well-being, while high life stress and job demands predicted negative well-being.

The differences in age, education level, marital status, daily work schedule, and job type contributed to the significant differences in work-related rumination. Older university logistics staff reported lower work-related rumination, possibly due to their reduced expectations for job advancement. Higher education levels are associated with increased work-related rumination, suggesting that those with higher education engage in more thoughtful considerations for their career development. Married university logistics staff tend to have higher levels of work-related rumination, which could be attributed to married employees having higher demands for job advancement as they need to support family life, leading to increased consideration of work-related issues. University logistics staff with flexible working hours exhibit higher levels of work-related rumination, possibly because they have the autonomy to choose their work hours and engage in post-work reflections on work. The increased and more frequent work-related rumination among technical employees (e.g., maintenance service personnel, procurement and inventory managers, and utility managers) compared to non-technical employees (e.g., dining hall staff and student housing administrators) may be attributed to the complex problem-solving, strategic planning, and critical decision-making inherent in technical roles.

There were significant differences in work engagement among university logistics staff from different occupational levels. In line with a previous study on hotel staff [49], mid-level and senior-level managers are more absorbed in their work compared to first-line staff and first-line managers. Higher-level managers, compared to those in first-line positions, are more likely to align internal career needs with external opportunities during the exploration stage, contributing to an engaged work state.

The aforementioned predictors for well-being offer essential insights for identifying effective strategies to enhance the well-being of university logistics staff. On the one hand, the university logistics department should improve the working conditions for employees. For example, it can adjust human resources rationally to alleviate the work pressure and reduce the workload of university logistics staff. Improving the work environment by minimizing noise, vibration, smoke, and dust is also essential. Additionally, increasing salaries to ensure a better quality of life materially can fundamentally enhance employee well-being. On the other hand, attention should be given to the mental health of university logistics staff, especially first-line staff. Providing psychological counselling services for university logistics staff is crucial, treating their mental health with the same concern as given to students and teachers in universities.

The strategies for enhancing well-being may be applicable in other professions as well. For example, chefs, who are also constantly exposed to smoke and experience the highest stress and burnout in the entire hospitality industry, benefit from a degree of financial security that ensures an acceptable standard of living, ultimately enhancing their well-being [50]. Additionally, construction workers who work in poor environments, with long hours, overloading, and high levels of physical effort also emphasize the importance of the impact of the quality of the physical work environment, fair and adequate salary, workload, and job support on their well-being. They also mentioned the role of perceiving others’ care for them in their well-being, which psychological counselling services can provide for them [51].

### 4.1. Strengths and Limitations

The present study marks the inaugural exploration into the well-being, work-related rumination, and work engagement of university logistics staff with diverse job types, employing SWELL, WRRQ, and UWES-9 assessments. The use of these well-established and validated scales enhances the precision and depth of our measurements, collectively providing a holistic view of the psychological landscape within the context of university logistics. The findings contribute to an enhanced comprehension of predictors of well-being and its associations with work-related rumination and work engagement in university logistics staff, providing a reference for later research. Furthermore, it sheds light on the often-overlooked well-being of logistics staff, bringing attention to a previously neglected aspect of research.

Although improvements in the workload, working hours, and remuneration can solve issues pertaining to overloading stress, ruminations about work, and low levels of well-being, the implementation of such changes might be challenging considering the sophisticated social and cultural contexts in some private and public institutions. In the situation of the latter, the results of the present study provide a perspective for authorities to improve the well-being of the logistics staff via indirect methods. For instance, support groups and activities can be arranged to improve the thinking style or overall attitudes that the logistics staff hold towards life and work. In addition, cognitive behaviour therapies could be beneficial for them to cope with the stress and build strategies to disconnect work from life during non-working hours.

However, the study had several limitations. On the one hand, the findings presented here are cross-sectional due to the nature of the research question under investigation. It was therefore not the focus of the present study to investigate claims of causality between well-being, work-related rumination and work engagement. Additional studies are necessary to further explore these questions of causality by using a longitudinal design which would enable researchers to track changes over time, offering a more nuanced understanding of how variations in well-being, work-related rumination, and work engagement reciprocally influence each other. This approach would not only enhance the robustness of our findings but also shed light on the potential mechanisms through which these variables interact and evolve throughout the university logistics staff’s professional lives. On the other hand, the present study did not confirm the effect of intervention strategies to enhance the well-being of university logistics staff.

### 4.2. Future Studies

Future studies can be further improved based on this experiment. First, semi-structured interviews could be conducted with university logistics staff in future studies to delve deeper into their psychological well-being and the underlying factors. Second, future researchers can validate the feasibility and effectiveness of different intervention methods like mindfulness-based stress reduction (MBSR) to enhance their well-being. Mindfulness-based programs are considered a promising way to reduce perceived stress and enhance well-being in workplaces [52]. While the effectiveness of MBSR has been demonstrated in various groups such as nurses, healthcare professionals, and college students [53,54,55,56], there is currently no research exploring the intervention effects on university logistics personnel and its role in enhancing their well-being. Therefore, future research can centre its attention on assessing the effectiveness of MBSR or other intervention methods within university logistics staff, exploring helpful ways to improve their well-being.

## 5. Conclusions

This study examined positive well-being, work-related rumination, and work engagement using SWELL, WRRQ, and UWES-9 and identified the factors affecting these among Chinese university logistics staff. The results confirm that positive characteristics and work engagement are predictors of positive well-being, both within and outside of work, whereas the presence of negative job characteristics and work-related rumination serve as predictors of low well-being.

## Figures and Tables

**Table 1 behavsci-14-00065-t001:** Descriptive statistics for participants’ demographic information.

Factor	Frequency	Percentage
Education level		
High school or lower education	31	11%
Associate degree	97	34.4%
Bachelor’s degree	153	54.3%
Master’s degree	1	0.4%
Marital status		
Married	253	89.7%
Unmarried	26	9.2%
Widowed	2	0.7%
Divorced	1	0.4%
Number of children		
No children	30	10.6%
One child	193	68.4%
Two children	53	18.8%
Three or more children	6	2.1%
Employment status		
Full-time	276	97.9%
Part-time	2	0.7%
Intern	4	1.4%
Daily work schedules		
Fixed schedule	250	88.7%
Shift work	21	7.4%
Flexible work hours	11	3.9%
Job types		
Dining hall staff	38	13.5%
Student housing administrators	50	17.7%
Maintenance service personnel	29	10.3%
Procurement and inventory managers	53	18.8%
Property managers	39	13.8%
Utilities managers	33	11.7%
Sanitation and pest control officers	8	2.8%
Landscaping and greenery personnel	9	3.2%
Facility managers	22	7.8%
Transportation department staff	1	0.4%
Occupational level		
First-line staff	201	71.3%
First-line manager	76	27%
Mid-level manager	4	1.4%
Senior-level manager	1	0.4%

**Table 2 behavsci-14-00065-t002:** Descriptive statistics for each variable in the SWELL questionnaire.

Variable	Frequency above Threshold	% above Threshold	Mean	SD	Skewness	Kurtosis
Health-related behaviours	214	75.40%	7.37	2.82	−0.385	−1.118
Personality	209	74.10%	7.31	3.01	−0.555	−1.007
Life satisfaction	205	72.70%	6.97	2.79	−0.73	−0.891
Life stress	87	30.90%	4.56	3.06	−1.317	0.651
Happiness	196	69.50%	6.88	2.94	−1.014	−0.836
Anxious/Depressed	89	31.60%	4.37	3.16	−1.346	0.678
Musculo-skeletal problems	63	22.30%	3.63	2.65	−0.474	0.893
Noise and vibration	128	45.40%	5.27	3.07	−1.653	0.09
Fumes	95	33.70%	4.51	3	−1.339	0.443
Job demands	128	45.40%	5.5	2.9	−1.621	0.014
Job control and support	146	51.80%	5.87	3.25	−1.773	−0.182
Perceived stress at work	124	44.00%	5.28	3.17	−1.722	0.095
Job satisfaction	144	51.10%	5.71	2.95	−1.558	−0.122
Physical and mental fatigue	136	38.20%	5.43	3.06	−1.693	−0.021
Efficiency at work	170	60.30%	6.28	2.94	−1.438	−0.442
Work–life balance	105	37.20%	4.71	2.87	−1.419	0.386
Happy at work	158	52.50%	6.07	3.28	−1.745	−0.267
Anxious/Depressed because of work	122	43.30%	5.2	3.21	−1.764	0.118

**Table 3 behavsci-14-00065-t003:** The scores of Work-Related Rumination Questionnaire and Utrecht Work Engagement Scale.

Item	Score ($\bar{x}$ ± *s*)	Skewness	Kurtosis	Cronbach’s α
Work-Related Rumination Questionnaire	28.73 ± 4.80	−1.128	5.409	0.233
Affective rumination dimension	14.43 ± 6.79	0.169	−1.724	0.967
Problem-solving pondering dimension	14.31 ± 5.59	−0.009	−1.526	0.939
Utrecht Work Engagement Scale	36.21 ± 12.51	−0.221	−0.834	0.956
Vigor dimension	12.65 ± 4.93	−0.238	−1.244	0.938
Dedication dimension	10.99 ± 4.16	0.042	−0.571	0.877
Absorption dimension	12.57 ± 4.08	−0.264	−0.238	0.847

**Table 4 behavsci-14-00065-t004:** Bivariate correlations between observed variables.

Variables	1	2	3	4	5	6	7	8	9	10	11	12	13	14	15	16	17	18	19	20
1. Health-related behaviours	1																			
2. Personality	0.887 **	1																		
3. Life satisfaction	0.840 **	0.888 **	1																	
4. Life stress	−0.677 **	−0.729 **	−0.665 **	1																
5. Happiness	0.806 **	0.809 **	0.856 **	−0.646 **	1															
6. Anxious/Depressed	−0.728 **	−0.768 **	−0.731 **	0.853 **	−0.671 **	1														
7. Musculo-skeletal problems	−0.469 **	−0.436 **	−0.451 **	0.602 **	−0.469 **	0.642 **	1													
8. Noise and vibration	−0.381 **	−0.379 **	−0.391 **	0.386 **	−0.410 **	0.410 **	0.431 **	1												
9. Fumes	−0.320 **	−0.306 **	−0.363 **	0.259 **	−0.309 **	0.325 **	0.316 **	0.647 **	1											
10. Job demands	−0.346 **	−0.373 **	−0.382 **	0.391 **	−0.358 **	0.437 **	0.375 **	0.757 **	0.631 **	1										
11. Job control and support	0.511 **	0.512 **	0.461 **	−0.363 **	0.451 **	−0.353 **	−0.222 **	−0.652 **	−0.487 **	−0.647 **	1									
12. Perceived stress at work	−0.414 **	−0.452 **	−0.452 **	0.481 **	−0.425 **	0.469 **	0.408 **	0.801 **	0.666 **	0.875 **	−0.767 **	1								
13. Job satisfaction	0.499 **	0.473 **	0.531 **	−0.239 **	0.496 **	−0.305 **	−0.248 **	−0.662 **	−0.566 **	−0.696 **	0.789 **	−0.754 **	1							
14. Physical and mental fatigue	−0.364 **	−0.390 **	−0.338 **	0.452 **	−0.356 **	0.452 **	0.388 **	0.729 **	0.566 **	0.824 **	−0.717 **	0.858 **	−0.655 **	1						
15. Efficiency at work	0.543 **	0.560 **	0.553 **	−0.313 **	0.514 **	−0.400 **	−0.243 **	−0.631 **	−0.497 **	−0.634 **	0.779 **	−0.694 **	0.800 **	−0.660 **	1					
16. Work–life balance	−0.314 **	−0.337 **	−0.303 **	0.366 **	−0.320 **	0.440 **	0.341 **	0.557 **	0.486 **	0.690 **	−0.590 **	0.682 **	−0.520 **	0.673 **	−0.533 **	1				
17. Happiness at work	0.504 **	0.488 **	0.535 **	−0.319 **	0.498 **	−0.403 **	−0.293 **	−0.733 **	−0.601 **	−0.787 **	0.823 **	−0.835 **	0.903 **	−0.756 **	0.823 **	−0.628 **	1			
18. Anxious/Depressed because of work	−0.431 **	−0.480 **	−0.446 **	0.441 **	−0.404 **	0.488 **	0.350 **	0.748 **	0.599 **	0.853 **	−0.759 **	0.885 **	−0.757 **	0.838 **	−0.746 **	0.708 **	−0.848 **	1		
19. Work-related rumination	−0.147 *	−0.146 *	−0.238 **	0.038	−0.196 **	0.130 *	0.102	0.297 **	0.307 **	0.340 **	−0.183 **	0.390 **	−0.365 **	0.267 **	−0.242 **	0.254 **	−0.360 **	0.389 **	1	
20. Work engagement	0.455 **	0.451 **	0.440 **	−0.319 **	0.423 **	−0.389 **	−0.239 **	−0.548 **	−0.417 **	−0.584 **	0.730 **	−0.635 **	0.691 **	−0.562 **	0.695 **	−0.502 **	0.736 **	−0.698 **	−0.032	1

* *p* < 0.05, ** *p* < 0.001.

**Table 5 behavsci-14-00065-t005:** Difference in work-related rumination and work engagement between university logistics staffs with different factors.

Factors	n	The Score of UWES-9	*t/F*	*p*	The Score of WRRQ	*t/F*	*p*
Age (ANOVA)							
16~25	15	38.27 ± 13.52	0.213	0.808	32.33 ± 8.13	17.711	<0.001 **
26~50	253	36.09 ± 12.60			28.86 ± 3.97		
51~76	14	36.21 ± 10.10			22.57 ± 8.07		
Gender (*t*-test)							
Male	138	35.14 ± 13.60	1.406	0.161	29.07 ± 4.75	1.159	0.247
Female	144	37.24 ± 11.32			28.41 ± 4.85		
Education level (ANOVA)							
High school or lower education	31	33.81 ± 14.78	0.711	0.492	23.03 ± 8.58	31.185	<0.001 **
Associate degree	97	35.98 ± 12.58			28.88 ± 3.91		
Bachelor’s degree	153	36.71 ± 11.91			29.81 ± 3.26		
Marital status (*t*-test)							
Unmarried	29	35.48 ± 16.82	−0.253	0.802	25.52 ± 8.94	−2.136	0.041 *
Married	253	36.30 ± 11.96			29.10 ± 3.94		
Number of children raised (ANOVA)							
No children	30	35.03 ± 16.69	0.576	0.563	27.67 ± 9.32	1.079	0.341
One child	193	36.76 ± 11.79			28.98 ± 3.77		
Two or more children	59	35.03 ± 12.47			28.47 ± 4.50		
Daily work schedule (ANOVA)							
Fixed schedule	250	35.80 ± 12.43	1.927	0.147	28.86 ± 4.08	7.83	<0.001 **
Work in shift	11	43.09 ± 17.72			32.00 ± 10.32		
Flexible work hours	21	37.52 ± 9.29			25.48 ± 6.80		
Job type (ANOVA)							
Dining hall staff	38	35.87 ± 14.61	0.865	0.535	26.37 ± 9.47	2.542	0.015 *
Student housing administrator	50	37.34 ± 12.52			28.02 ± 4.47		
Maintenance ser vice personnel	29	35.55 ± 13.59			29.55 ± 3.19		
Procurement and inventory manager	53	34.38 ± 11.49			29.77 ± 1.91		
Property manager	39	35.31 ± 12.92			28.49 ± 4.38		
Utilities manager	33	39.88 ± 10.48			29.76 ± 1.75		
Facility manager	22	33.82 ± 11.48			30.09 ± 4.07		
Other job types (Transportation department staff, Sanitation and pest control officer, and Landscaping and greenery personnel)	18	38.44 ± 12.78			28.33 ± 2.89		
Occupational level (ANOVA)							
First-line staff	201	36.11 ± 12.25	4.007	0.019 *	28.60 ± 5.35	0.393	0.675
First-line manager	76	35.47 ± 12.96			29.13 ± 2.79		
Mid-level and Senior-level manager	5	51.60 ± 5.18			28.00 ± 6.36		

* *p* < 0.05, ** *p* < 0.001.

**Table 6 behavsci-14-00065-t006:** Post hoc comparison of factors with significant differences.

Dependent Variable	Factors (I)	Factors (J)	Mean Difference (I-J)	Std. Error	*p*	95%Confidence Interval	
						Lower Bound	Upper Bound
Work-related rumination	Age						
	16~25	26~50	3.47 *	1.20	0.017	0.502	6.440
		51~76	9.76 **	1.68	<0.001	5.609	13.913
	26~50	16~25	−3.47 *	1.20	0.017	−6.440	−0.502
		51~76	6.29 **	1.24	<0.001	3.222	9.357
	51~76	16~25	−9.76 **	1.68	<0.001	−13.913	−5.609
		26~50	−6.29 **	1.24	<0.001	−9.357	−3.222
	Education level						
	High school or lower education	Associate degree	−5.84 **	0.90	<0.001	−8.059	−3.628
		Bachelor’s degree	−6.77 **	0.85	<0.001	−8.892	−4.663
	Associate degree	High school or lower education	5.84 **	0.90	<0.001	3.628	8.059
		Bachelor’s degree	−0.93	0.56	0.258	−2.327	0.459
	Bachelor’s degree	High school or lower education	6.77 **	0.85	<0.001	4.663	8.892
		Associate degree	0.93	0.56	0.258	−0.459	2.327
	Daily work schedule						
	Fixed schedule	Work in shift	−3.13	1.44	0.097	−6.691	0.419
		Flexible work hours	3.38 *	1.06	0.007	0.765	6.010
	Work in shift	Fixed schedule	3.13	1.44	0.097	−0.419	6.691
		Flexible work hours	6.52 *	1.74	0.001	2.228	10.819
	Flexible work hours	Fixed schedule	−3.38 *	1.06	0.007	−6.010	−0.765
		Work in shift	−6.52 *	1.74	0.001	−10.819	−2.228
Work engagement	Occupational level						
	First-line staff	First-line manager	0.63	1.66	0.930	−3.465	4.737
		Mid-level and Senior-level manager	−15.49 *	5.60	0.023	−29.280	−1.701
	First-line manager	First-line staff	−0.63	1.66	0.930	−4.737	3.465
		Mid-level and Senior-level manager	−16.12 *	5.71	0.020	−30.188	−2.064
	Mid-level and Senior-level manager	First-line staff	15.49 *	5.60	0.023	1.701	29.280
		First-line manager	16.12 *	5.71	0.020	2.064	30.188

* *p* < 0.05, ** *p* < 0.001.

**Table 7 behavsci-14-00065-t007:** Odds ratio of each IV on life satisfaction.

Variables	Odds Ratio	95% C.I for Odds Ratio	*p*
Health-related behaviours (healthy)	6.391	[1.196, 34.140]	0.030 *
Personality (positive)	20.126	[4.125, 98.193]	<0.001 **
Life stress (low)	14.087	[1.184, 167.672]	0.036 *
Anxious/Depressed (low)	0.060	[0.017, 0.208]	<0.001 **
Musculo-skeletal problems (low)	0.970	[0.142, 6.635]	0.975
Noise and vibration (low)	0.895	[0.047, 17.145]	0.941
Fumes (low)	2.568	[0.168, 39.313]	0.498
Job demands (low)	10.202	[0.405, 256.741]	0.158
Job control and support (high)	0.127	[0.006, 2.59]	0.180
Perceived stress at work (low)	0.188	[0.019, 1.833]	0.150
Physical and mental fatigue (low)	2.375	[0.099, 56.738]	0.593
Efficiency at work (high)	0.889	[0.776, 1.019]	0.092
Work–life balance (high)	1.021	[0.941, 1.108]	0.615
Anxious/Depressed because of work (low)	0.014	[0.001, 0.188]	0.001 *
Work-related rumination (low)	0.889	[0.776, 1.019]	0.092
Work engagement (high)	1.021	[0.941, 1.108]	0.615

* *p* < 0.05, ** *p* < 0.001.

**Table 8 behavsci-14-00065-t008:** Odds ratio of each IV on happiness.

Variables	Odds Ratio	95% C.I for Odds Ratio	*p*
Health-related behaviours (healthy)	0.070	[0.011, 0.443]	0.005 *
Personality (positive)	0.109	[0.017, 0.683]	0.018 *
Life stress (low)	0.061	[0.019, 0.196]	<0.001 **
Anxious/Depressed (low)	0.723	[0.13, 4.032]	0.711
Musculo-skeletal problems (low)	0.247	[0.055, 1.104]	0.067
Noise and vibration (low)	0.782	[0.094, 6.496]	0.820
Fumes (low)	5.297	[0.845, 33.197]	0.075
Job demands (low)	7.829	[0.614, 99.826]	0.113
Job control and support (high)	1.591	[0.28, 9.048]	0.601
Perceived stress at work (low)	0.131	[0.029, 0.593]	0.008 *
Physical and mental fatigue (low)	0.145	[0.02, 1.045]	0.055
Efficiency at work (high)	4.019	[1.568, 10.300]	0.004 *
Work–life balance (high)	1.040	[0.979, 1.104]	0.203
Anxious/Depressed because of work (low)	0.109	[0.017, 0.683]	0.018
Work-related rumination (low)	0.896	[0.808, 0.994]	0.037 *
Work engagement (high)	1.040	[0.979, 1.104]	0.203

* *p* < 0.05, ** *p* < 0.001.

**Table 9 behavsci-14-00065-t009:** Odds ratio of each IV on job satisfaction.

Variables	Odds Ratio	95% C.I for Odds Ratio	*p*
Health-related behaviours (healthy)	0.313	[0.056, 1.749]	0.186
Personality (positive)	0.496	[0.071, 3.457]	0.479
Life stress (low)	0.158	[0.022, 1.146]	0.068
Anxious/Depressed (low)	3.513	[1.106, 11.162]	0.033 *
Musculo-skeletal problems (low)	1.365	[0.418, 4.463]	0.606
Noise and vibration (low)	2.456	[0.561, 10.747]	0.233
Fumes (low)	0.377	[0.107, 1.333]	0.130
Job demands (low)	0.255	[0.095, 0.683]	0.007 *
Job control and support (high)	3.653	[1.396, 9.559]	0.008 *
Perceived stress at work (low)	0.281	[0.075, 1.058]	0.061
Physical and mental fatigue (low)	2.252	[0.419, 12.091]	0.344
Efficiency at work (high)	5.374	[1.849, 15.617]	0.002 *
Work–life balance (high)	1.073	[1.021, 1.127]	0.005 *
Anxious/Depressed because of work (low)	0.496	[0.071, 3.457]	0.479
Work-related rumination (low)	0.889	[0.813, 0.971]	0.009 *
Work engagement (high)	1.075	[1.028, 1.124]	0.002 *

* *p* < 0.05.

**Table 10 behavsci-14-00065-t010:** Odds ratio of each IV on happy at work.

Variables	Odds Ratio	95% C.I for Odds Ratio	*p*
Health-related behaviours (healthy)	2.226	[0.160, 30.889]	0.551
Personality (positive)	8.027	[0.421, 152.983]	0.166
Life stress (low)	10.700	[0.922, 124.21]	0.058
Anxious/Depressed (low)	0.321	[0.027, 3.770]	0.366
Musculo-skeletal problems (low)	10.086	[0.934, 108.928]	0.057
Noise and vibration (low)	19.616	[1.721, 223.571]	0.017
Fumes (low)	0.008	[0.001, 0.100]	<0.001 **
Job demands (low)	0.040	[0.012, 0.137]	<0.001 **
Job control and support (high)	13.472	[3.787, 47.921]	<0.001 **
Perceived stress at work (low)	0.013	[0.001, 0.182]	0.001 **
Physical and mental fatigue (low)	5.898	[0.389, 89.368]	0.201
Efficiency at work (high)	9.725	[2.449, 38.621]	0.001 *
Work–life balance (high)	1.062	[0.993, 1.135]	0.078
Anxious/Depressed because of work (low)	8.027	[0.421, 152.983]	0.166
Work-related rumination (low)	0.578	[0.064, 5.262]	0.627
Work engagement (high)	1.060	[1.011, 1.111]	0.015 *

* *p* < 0.05, ** *p* < 0.001.

## Data Availability

Data are contained within the article.

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
