# Peer review of "Positive Well-Being, Work-Related Rumination and Work Engagement among Chinese University Logistics Staff"

_behavsci, 2024, doi:10.3390/bs14010065_

Round 1

Reviewer 1 Report

Comments and Suggestions for Authors

 1. Please add more keywords (up to 10) for wider indexing.

2. Please present an aim of your study as well as hypotheses clearly.

3. Procedure should be described before Measures and Participants. You plan a procedure and then you recruit participants, collect and analyze data, etc.

4. Please add references for Chinese versions of questionnaires used. Indicate that Chinese versions were used.

5. "The Smith Well-being Questionnaire (SWELL), a single-item scale, was adopted to assess the well-being of the participants [18].". This is not a single-item scale as it consists of 19 items (as the authors stated). Please reconsider the description. Here is: One item for one characteristic of well-being.

6. What do the higher scores in SWELL mean? For example, stress. Low stress? High stress? And other 19 factors. SWELL variables are your key variables, please describe them in detail.

7. Please calculate internal consistency reliability for your questionnaires in this study.

8. Table 3 and 4 can be merged. Also, please add descriptive statistics for your SWELL variables here. Please add M, SD, skewness, kurtosis, and alpha Cronbach's here. Descriptive statistics should be presented.

9. Table 5 and related analyses: Please clearly indicate where you applied ANOVA and where t-tests. Moreover, some groups are extremely small (e.g., n =1 in occupational level or job type), therefore, the correctness of these analyses has many concerns. Therefore, please recalculate according to requirements of using these tests. Did you use post-hoc tests in ANOVA? And What?

10. Table 6 is placed in an appropriate place in the manuscript.

11. Did you present only significant predictors in regression analyses? Please show all added predictors with their p-values. It is unclear what predictors were input in the model. In general, statistical analyses should be presented correctly. There are many problems with presentations. Replicability of this paper is very low, as it is unclear what the authors did here.

12. "Therefore, a brief survey was conducted in the present study," - this is tautology. Please re-read your paper and reconsider some similar sentences with tautology.

13. Please indicate practical implications.    

Reviewer 2 Report

Comments and Suggestions for Authors

The paper addresses an important and relevant topic, exploring the relationship between positive well-being, work-related rumination, and work engagement within the context of university logistics staff. However, there are some major issues that need to be addressed before the paper is ready to be published.

The abstract needs to be rewritten in its entirety, it is lacking vital information about the study, how it was conducted, the number of respondents etc.

The paper pertains to a situation in present day Chine, which should be noted in the title, the abstract and in the introduction. Most universities outside Chine would not regard ideological and political work as a responsibility for their staff.

Why is the group university logistics staff interesting to investigate? The paper states that this group has not been studied before, but the absence of a previous study is not an argument.  There need to be  a clear argument in what way information about this group may contribute to the accumulated knowledge regarding well-being and health.

The work situation of the university logistics staff needs to be described in somewhat more detail. What do they do exactly and in what way is their work cognitively and emotionally demanding?

Novelty: the paper fails to bring any novelty. It is not enough to state that the present group has not been studied before. The research objectives must also communicate the intended scholarly contribution of the research, which might be: (1) theoretical (developing or contributing to new theory or testing existing theory), (2) methodological (developing or contributing to new methods); or (3) empirical (new applications of existing methods or theories, or new types of evidence).

The data presented in the article may be interesting to scholars, but the authors need to develop their argument and a clearer rationale for the study. They need to argue why the study is important and how the results may contribute to knowledge.

Comments on the Quality of English Language

English language use needs to be checked. 

Round 2

Reviewer 1 Report

Comments and Suggestions for Authors

The paper was improved satisfactorily.

Minor comments:

Table 4: Please indicate what asterisks mean. * p < 0.05? ** p < 0.01?

Regression analyses: Please indicate clearly how many predictors were input in the model and how many models were here. Were the predictors input in the model together (and at the same time) or you present separate analysis for individual predictors? See my previous comments on regression analyses.

P-value can not equal 0.000. Please replace by p < 0.001.
